# Stress-Relieving Effects of Sesame Oil Aroma and Identification of the Active Components

**DOI:** 10.3390/molecules27092661

**Published:** 2022-04-20

**Authors:** Hiroaki Takemoto, Yuki Saito, Kei Misumi, Masaki Nagasaki, Yoshinori Masuo

**Affiliations:** 1Department of Molecular Toxicology, Faculty of Pharmaceutical Sciences, Toho University, 2-2-1 Miyama, Funabashi, Chiba 274-8510, Japan; 2Laboratory of Neuroscience, Department of Biology, Faculty of Science, Toho University, 2-2-1 Miyama, Funabashi, Chiba 274-8510, Japan; 5218033s@st.toho-u.jp (Y.S.); 5218077m@st.toho-u.jp (K.M.); 5218056n@st.toho-u.jp (M.N.)

**Keywords:** sesame oil aroma, 2,5-dimethylpyrazine, 2-methoxy phenol, water-immersion stress, antianxiety, elevated plus-maze test, dual specificity phosphatase 1

## Abstract

(1) Sesame oil aroma has stress-relieving properties, but there is little information on its effective use and active ingredients. (2) Methods: ICR male mice were housed under water-immersion stress for 24 h. Then, the scent of sesame oil or a typical ingredient was inhaled to the stress groups for 30, 60, or 90 min. We investigated the effects of sesame oil aroma on mice behavior and the expression of the dual specificity phosphatase 1 (*DUSP1*) gene, a candidate stress marker gene in the brain. (3) Results: In an elevated plus-maze test, the rate of entering into the open arm of a maze and the staying time were increased to a maximum after 60 min of inhalation, but these effects decreased 90 min after inhalation. As for the single component, anxiolytic effects were observed in the 2,5-dimethylpyrazine and 2-methoxy phenol group, but the effect was weakened in the furfuryl mercaptan group. The expression levels of *DUSP1* in the hippocampus and striatum were significantly decreased in 2,5-dimethylpyrazine and 2-methoxy phenol groups. (4) Conclusions: We clarified the active ingredients and optimal concentrations of sesame oil for its sedative effect. In particular, 2,5-dimethylpyrazine and 2-methoxy phenol significantly suppressed the stress-induced changes in the expression of *DUSP1*, which are strong anti-stress agents. Our results suggest that these molecules may be powerful anti-stress agents.

## 1. Introduction

Central nervous system disorders have a great impact on society due to the general aging process of the population and lifestyle factors. Stress is one of the most prevalent psychological disorders in developed countries and leads to other clinical concerns, such as anxiety, insomnia, or depression [1]. Prolonged periods of stress produce deleterious effects that might become chronic and/or life-threatening. These effects are related to its involvement in medical conditions such as metabolic syndrome, cardiovascular disease, type 2 diabetes mellitus, allergies, and autoimmune diseases. Furthermore, brain disorders and functions are notably influenced by stress. Its role in precipitating psychiatric pathologies is either suggested or demonstrated for conditions such as mood disorders, anxiety disorders, and post-traumatic stress disorder [2].

Benzodiazepines (BZD) and selective serotonin reuptake inhibitors (SSRIs) are highly prescribed as anxiolytic and antidepressant drugs, respectively. BZD, such as diazepam, lorazepam, or alprazolam, produces calming effects via binding to GABA_A_ receptors but may also produce somnolence, cognitive impairment, and ataxia as adverse drug reactions [3]. SSRIs (e.g., fluoxetine, paroxetine, and citalopram) are prescribed as antidepressants because they are able to selectively block the serotonin transporter, but side effects include sexual dysfunction, sleep disturbance, and suicidal tendencies [4]. Both groups of medicines can produce withdrawal and rebound effects when their use discontinues.

Phytotherapy has recently attracted much attention for reducing daily stress and preventing central disorders. Particularly, essential oils have a long tradition in pharmaceutical sciences as natural products with pharmacological, cosmetic, and nutritional applications [5]. The use of essential oil in the form of aromatherapy is widely extended, with some of them being used as agents to relieve anxiety and stress [6]. For example, lavender essential oil is traditionally used and approved by the European Medicines Agency as herbal medicine to relieve stress and anxiety [1].

Previously, we investigated the anti-stress effect of sesame oil aroma [7]. Various kinds of foods have drawn attention in recent years due to their stress relaxation effects. Sesame seeds have been a popular healthy food product in Japan since ancient times, and Japan imports the second-largest amount as a nation, following the US and Europe [8]. Sesame lignans, which show antioxidant effects, are known as the most popular active ingredient in sesame, while sesame seeds have a characteristic aroma. When mice were subjected to 24 h of water-immersion stress as a sleep-disordered stress, the number of entries into the open arm of the maze decreased, and anxiety-like behavior was observed in the elevated plus-maze test. However, the inhalation of sesame oil aroma for 90 min after water-immersion stress attenuated the anxiety-like behavior.

Furthermore, the expression of *DUSP1* (dual specificity phosphatase 1) gene was investigated. *DUSP* is known to be negative regulators of the mitogen-activated protein kinase cascade and modulate diverse neural functions, such as neurogenesis, differentiation, and apoptosis. *DUSP* genes have furthermore been associated with mental disorders such as depression and neurological disorders [9]. Although the expression of *DUSP1* in the brain was increased by water-immersion stress, the increased level was significantly attenuated by the sesame oil aroma.

The unique fragrance ingredients of roasted sesame seeds are composed of several molecules. Xiao Jia et al. reported that fifteen volatile compounds with the highest odor activity values were selected as the key odors contributing to the flavor profile of sesame oil aroma, including 2-methyl-propanal, 2-methyl-butanal, furaneol, 1-octen-3-one, 4-methyl-3-penten-2-one, 1-nonanol, 2-methyl-phenol, 2-methoxy phenol, 2-methoxy-4-vinylphenol, 2,5-dimethylpyrazine, 2-furfuryl mercaptan, 2-thiophenemethanethiol, methanethiol, methional, and dimethyl trisulfide [10]. Generally, sulfur-containing compounds, pyrroles, and pyrazines seem to play important roles in the characteristic odor of sesame oil. Furthermore, among these compounds, pyrazines are major volatiles in sesame oil [11]. They are commonly found in the aroma of roasted coffee [12], and it has been reported that coffee bean volatiles have anxiolytic and hypnotic effects in mice [13].

In the present study, we analyzed the effect of sesame oil aroma in more detail. Fragrance inhalation has primary effects and toxic effects, so the optimal duration of inhalation administration was determined. Furthermore, we identified the active components of sesame oil aroma. Previously, pyrazines, methoxyphenols, and sulfur compounds were detected from sesame oil aroma by GC-MS analysis. We investigated the anxiolytic effect of 2,5-dimethylpyrazine, 2-methoxy phenol, and furfuryl mercaptan (Figure 1). In this study, to obtain the data for the effective use of sesame oil scent, an analysis was conducted of the anti-stress effects of sesame oil at different inhalation times, and the effects of typical compounds contained in sesame oil were evaluated from the elevated plus-maze test and analysis of genetic variation by real-time PCR.

## 2. Results

### 2.1. Anxiolytic Effect of Sesame Oil Aroma

After 24 h of stress-loading, the mice inhaled sesame oil aroma for 0 min, 30 min, 60 min, or 90 min. In an elevated plus-maze test, the number of entries into each of the open arms of the maze and the closed arm was measured, and a ratio of the open arm entry number per entry number to the open and closed arms was calculated as the open arm entry rate (Figure 2A). Figure 2B shows the staying time in the open arm. The average value of the open arm entry rate of the 0 min group was 23.6%, while it was 65.8% for the 30 min group and 90.2% for the 60 min group (Figure 2A). The ratio of the open arm entry increase was dependent on the inhalation time. However, the 90 min group value was 62.8%, indicating a decrease in the anxiolytic effect. The result of the staying time in the open arm was similar to that of the open arm entry rate. The average value of the staying time in the open arm of the 0 min group was 128.8 s, while it was 312.7 s for the 30 min group and 391.0 s for the 60 min group (Figure 2B). However, the value in the 90 min group was 253.3 s. These results suggest that the odor of sesame oil may have an anxiety-depressing action. A significant stress-reducing effect was found to appear 30 min after inhalation administration, and it was found that the greatest effect was observed 60 min after inhalation administration.

### 2.2. Anxiolytic Effect of Constituents of Sesame Oil Aroma

After 24 h of stress-loading, the mice inhaled 2,5-dimethylpyrazine, 2-methoxy phenol, or furfuryl mercaptan, a constituent of sesame oil aroma, for 30 min. The open arm entry rate and the staying time in the open arm are shown in Figure 3. The open arm entry rate of the sesame oil group was 65.8%, while that of the 2,5-dimethylpyrazine group was 83.3%, and that of the 2-methoxy phenol group was 88.2%. However, the effect of furfuryl mercaptan was weakened, and the open arm entry rate was only 22.6% (Figure 3A). Similarly, the staying time in the open arm of the 2,5-dimethylpyrazine (479.7 s) and 2-methoxy phenol groups (519.4 s) were longer than that of sesame oil (312.7 s), while that of the furfuryl mercaptan group (118.3 s) was significantly shorter than that of sesame oil (312.7 s) (Figure 3B). These results suggested that 2,5-dimethylpyrazine and 2-methoxy phenol are the active ingredients involved in the stress-relieving effect of sesame oil.

### 2.3. DUSP1 Expression after Sesame Oil Aroma Inhalation

After 24 h of stress-loading, the mice inhaled sesame oil aroma for 0 min, 30 min, 60 min, or 90 min. The expression levels of *DUSP1* gene were measured at the end of inhalation administration (Figure 4). The expression level of *DUSP1* in the hippocampus was significantly increased in the 30 min inhalation group (*p* < 0.05, two-way ANOVA). However, the expression levels in the 60 and 90 min inhalation groups decreased (Figure 4A). On the other hand, in the striatum, the expression level significantly decreased after 30 min and 60 min inhalation, and this tendency was also observed in 90 min inhalation group (Figure 4B).

### 2.4. DUSP1 Expression after Inhalation of Constituents of Sesame Oil Aroma

After 24 h of stress-loading, the mice inhaled 2,5-dimethylpyrazine, 2-methoxy phenol, or furfuryl mercaptan for 90 min (Figure 5). The expression level in both the striatum and hippocampus significantly decreased in the 2,5-dimethylpyrazine and 2-methoxy phenol groups (*p* < 0.05, two-way ANOVA). The effect of furfuryl mercaptan was lower than the other two compounds, and the expression level tended to decrease; however, there were no significant differences.

## 3. Discussion

We have previously reported that sesame oil aroma has an anti-stress effect [7]. In the present study, we performed a detailed analysis of the functionality of sesame oil aroma. Firstly, the effect of inhalation time on the anti-stress effect was analyzed. To evaluate the effect of anti-stress on behavior, we conducted an elevated plus-maze test. The water-immersion stress has been shown to reduce the penetration rate to the open arm and the staying time in the open arms [7]. The anti-stress effect of sesame oil aroma appeared 30 min after inhalation, and the effect was even stronger 60 min after inhalation (Figure 2). On the other hand, the anti-stress effect weakened after 90 min of inhalation. It is known that the inhaled volatile constituents act on the central nervous system through olfactory nerves and the bloodstream [14]. It was suggested that the increased blood levels of the active ingredient in sesame oil may cause toxicity after 90 min of inhalation.

Lavender oil mainly includes linalool and linalyl acetate, and it has been reported that linalool has a sedative effect on humans by vapor inhalation at low doses [15]. However, it also has nerve toxicity, and the LC_50_ for mammals is 2740 mg/m^3^ [16]. We previously investigated the relationship between blood concentration of fragrance components and onset of the sedative effect by intravenous administration test. Valerena-4,7(11)-diene, a component of *Valeriana officinalis* oil, was administered intravenously at doses of 10, 100, and 1000 µg/kg, and open field tests were performed. Sedative effects were greater in the 100 µg/kg group than in the 10 µg/kg group; however, treatment with 1000 µg/kg caused a decreasing effect, suggesting an overdose. It was suggested that prolonged inhalation of sesame oil flavor weakened anti-stress effects.

We then examined *DUSP1* gene expressions in the hippocampus and striatum. The hippocampus plays an important role in stress regulation. It exerts inhibitory control over hypothalamic-pituitary-adrenal-axis activity and is also more broadly involved in cognitive and affective processing via its widespread connections with other limbic prefrontal regions [17]. The striatum is classically described as playing a key role in motor function and is a part of the basal ganglia, which is important for adjusting the execution of motor habits. Thus, deficits in motor automaticity are a characteristic of basal ganglia-related illnesses, such as Parkinson’s disease.

Furthermore, anxiety, one of the main symptoms of depression, may be regulated by the striatum. As one of the striatum-related circuit mechanisms underlying anxiety, the striatal–prefrontal pathway is involved and becomes less connected, the cortico-striatal connections are also impaired, and anxiety is expressed [18]. It is known that the expression level of *DUSP1* is elevated in the rat brain experiencing chronic stress [19] and decreased by treatment with antidepressants [20]. *DUSP1* gene may be a representative of a promising new drug target for the treatment of depression and other mood disorders. The present study demonstrated that the scent of sesame oil significantly inhibited the expression level of *DUSP1* in the striatum. On the other hand, the level in the hippocampus was increased instantaneously in 30 min inhalation group (Figure 4).

In the present study, we further analyzed the anti-stress effects of typical aromatic components of sesame oil by inhalation administration. 2,5-Dimethylpyrazine and 2-methoxy phenol showed stronger anti-stress effects than sesame oil, while furfuryl mercaptan had no effect (Figure 3). Pyrazines and phenols are also commonly found in the roasted aroma of coffee and tea. These compounds have been suggested to have a stress-reducing effect [21] 2,5-Dimethylpyrazine 2-methoxy phenol, roasted aroma contained in sesame oil, seems to be an active ingredient involved in its stress-reducing effects. It is reported the effect of alkylpyrazine derivatives on pentobarbital-induced sleeping time, picrotoxicin-induced convulsion, and γ-aminobutyric acid levels in mouse brain [21]. These results suggest that alkylpyrazine derivatives may strengthen the GABAnergic system in the brain.

On the other hand, sulfur compounds such as furfuryl mercaptan have also been identified as the characteristic aroma component in roasted coffee, wheat bread, and popcorn, and these compounds were confirmed as key contributors to the flavor of sesame seeds [22]. Furfuryl mercaptan did not show anti-stress effects in the elevated plus-maze test, and the expression level of *DUSP1* tended to decrease but did not differ from the control. Unpleasant smelling gases such as halitosis are caused by volatile sulphur compounds such as hydrogen sulphide, methyl mercaptan, and dimethyl sulphide [23]. Furfuryl mercaptan may also exhibit anti-stress effects, although the concentration used in this experiment is considered to be the borderline concentration at which both pleasant and unpleasant effects appear. Therefore, it is necessary to verify the effect of low concentrations of furfuryl mercaptan in the future. *DUSP1* in the hippocampus was elevated 30 min after sesame oil administration, but this may be due to an unpleasant odor. However, the level of *DUSP1* decreased after 60 min of sesame oil administration, which may reflect the anti-stress effects of the compounds found in this study. The anti-stress effects of sesame oil may be maximized with 60 min of inhalation.

The safety of sesame oil has been tested for lignans [24]. A single-blind, placebo-controlled, parallel-group, and multiple oral dose study was conducted on 48 healthy subjects to investigate the pharmacokinetics and safety of multiple oral doses of sesame lignans (sesamin and episesamin). No serious adverse events were observed in this study. The pharmacokinetic study results demonstrate that no accumulation was observed following multiple 50 mg doses of sesame lignans. Therefore, the use of fragrances, especially for their anti-stress effects, is considered safe to use.

Recently, aromatherapy has attracted much attention as an alternative medicine, especially for psychosomatic diseases caused by stress. We have studied the effects of the odor of sesame oil and clarified the optimal time for inhalation of sesame oil, which has a sedative effect. Furthermore, we found that the sesame odor components, 2,5-dimethylpyrazine and 2-methoxy phenol, significantly suppressed stress-induced changes in the expression of *DUSP1*, a candidate stress marker gene in the striatum and hippocampus.

## 4. Materials and Methods

### 4.1. Materials

Aromatic sesame oil used in this study was purchased from Kuki Sangyo Co., Ltd. (Mie, Japan). This oil was produced by pressing and extracting white sesame seeds from Guatemala, followed by a cooling process. 2,5-Dimethylpyrazine (purity > 98%, GC), 2-methoxy phenol (purity > 98%, GC) and furfuryl mercaptan (purity > 98%, GC) were purchased from Tokyo Chemical Industry Co., Ltd. (Tokyo, Japan).

### 4.2. Animal Experiment

This study was carried out according to the animal experiment handling provisions of Toho University’s Animal Experiment Committee. The committee approved the animal experiment design plan. ICR male mice at 5 weeks of age purchased from Clea Japan (Tokyo, Japan) were housed in an acrylic cage, three animals per cage, and kept under the 12 h light/dark cycle (light period 08:00–20:00) at 24 ± 2 °C. Food and water were allowed ad libitum. After adapting to the rearing environment for over a week, mice were randomly divided into groups of four mice per group. All groups of mice were bred for 24 h in a cage with water to the extent to which the limb of the mouse is immersed (water depth of about 1 cm), whereby the mice experience a stress response due to the induced discomfort and insomnia [25]. The inhalation commenced at 10:30, just after 24 h of stress-loading. The odor was placed in a glass box in which a piece of filter paper was impregnated with 50 μL of sesame oil, or 5 μL of 2,5-dimethylpyrazine, 2-methoxy phenol, or furfuryl mercaptan was adhered to the inside, and mice inhaled for 30, 60, or 90 min. Physiological saline was used as a control. After completing the inhalation administration, behavioral tests and dissections were performed as follows. This study was carried out according to the animal experiment handling provision by Toho University Animal Experiment Committee. The animal experiment design plan has been approved by the committee (approval number: 21-31-476, date: 1 April 2021).

### 4.3. Behavioral Analysis

The evaluation of anxiety-like behavior was performed by the elevated plus-maze test for 10 min just after completion of the inhalation of the aroma. The maze consists of an arm length of 30 cm, an arm width of 5 cm, a height from the floor of 60 cm, and a closed arm wall height of 20 cm. Immediately after putting the mouse in the center of the maze, the behavior of the animals was recorded with a video camera. The number of entering the open arm and the staying time and the number of entering into the closed arm and the staying time, were automatically recorded using the behavioral tracking software ANY-maze (Muromachi Kikai, Tokyo, Japan) [7].

### 4.4. Total RNA Extraction

The mice were decapitated, and the whole brains were collected. The whole brain was dissected on ice, and the striatum and hippocampus were sampled. The brain tissue was immediately frozen in liquid nitrogen, and the frozen tissue was pulverized in liquid nitrogen to convert it into powder form. One milliliter of Qiazol (Qiagen, Hilden, Germany) was added to each brain tissue sample and stirred until the sample was completely dissolved. The total RNA was extracted with RNeasy Mini Kit (Qiagen) and RNase-Free DNase set (Qiagen) as per the manufacturer’s protocols. The RNA was dissolved in RNase-free water, and the concentration and the purity of the obtained total RNA were measured using NanoDrop (Thermo Fisher Scientific, Waltham, MA, USA). Samples with a low purity were ethanol-precipitated with Ethachinmate (Nippon Jean, Tokyo, Japan). The total RNA was stored at −80 °C [7].

### 4.5. cDNA Synthesis

Reverse transcription was performed from the obtained RNA to synthesize cDNA using ReverTra Ace^®^ qPCR RT Master Mix (Toyobo, Osaka, Japan). First, the amount of sample was calculated so that the RNA concentration of each sample was 1 pg/μL to 1 μg/μL in 10 μL. A total of 6 μL of nuclease-free water was added to 2 μL of each sample, and 2 μL of 2 × RT Master Mix (ReverTra Ace^®^ qPCR RT Master Mix, Toyobo) was added. cDNA was synthesized by reverse transcription using GeneAmp PCR System 9700 (Applied Biosystems, Thermo Fisher Scientific). The conditions for cDNA synthesis were 15 min at 37 °C, 5 min at 50 °C, and 5 min at 98 °C. The synthesized cDNA was stored at −20 °C [7].

### 4.6. Real-Time Reverse Transcription-Polymerase Chain Reaction (Real-Time RT-PCR)

For real-time RT-PCR, we used a TB Green Premix Ex Taq II (Takara Bio, Shiga, Japan). For each 2 μL of the cDNA sample, 1.6 μL of primer (Forward and Reverse, 0.8 μL each) (Table 1) (Takara Bio) was added, and 8.5 μL of RNase free water, 12.5 μL of TB Green Premix Ex Taq II and 0.4 μL of Rox Reference Dye was added to make a total of 25 μL. The expression level of cDNA was measured using an Applied Biosystems^®^ 7500 real-time PCR system (Thermo Fisher Scientific). The holding stage was at 50 °C for 2 min and 95 °C for 10 min. The cycling stage for 40 cycles was at 95 °C for 15 s, and 60 °C for 1 min. The melt curve stage was at 95 °C for 15 s, 60 °C for 1 min, 95 °C for 30 s, and 60 °C for 15 s. The analysis of the gene expression level of each sample was carried out in comparison with the expression level of glyceraldehyde 3-phosphate dehydrogenase (*GAPDH*), which is a housekeeping gene. The expression levels of *DUSP1* were calculated as correction values divided by the expression level of *GAPDH*. The primer sequences of each gene used in this study were the same as in the previous report [7].

### 4.7. Statistical Analysis

The results were expressed as mean ± SEM. ANOVA analysis was performed for the significance among four groups, and further analysis was conducted using the Tukey method. In some cases, the results were analyzed by a Student’s *t*-test to determine the significant difference between the two groups. *p* < 0.05 was considered to be statistically significant. Analyses were performed with BellCurve for Excel (Social Survey Research Information Co., Ltd. Tokyo, Japan).

## 5. Conclusions

The effect of sesame oil aroma and its associated typical ingredients on behavior and stress-related biomarkers was investigated. In an elevated plus-maze test, the anxiolytic effect of sesame oil aroma was observed to maximum after 60 min inhalation, but the effect was decreased 90 min after inhalation. As for the single component, anxiolytic effects were observed in the 2,5-dimethylpyrazine and 2-methoxy phenol groups, but the effect was weakened in the furfuryl mercaptan group. The expression of *DUSP1* also showed similar results to an elevated plus-maze test. We clarified the sesame oil’s active ingredients and optimal concentrations for its sedative effects. 2,5-Dimethylpyrazine and 2-methoxy phenol significantly suppressed the stress-induced changes in the expression of *DUSP1*, a candidate stress marker gene in the striatum and hippocampus.

## Figures and Tables

**Figure 1 molecules-27-02661-f001:**
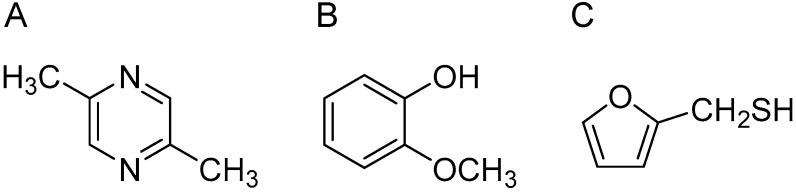
Chemical structures of 2,5-dimethylpyrazine (**A**), 2-methoxy phenol (**B**), and furfuryl mercaptan (**C**).

**Figure 2 molecules-27-02661-f002:**
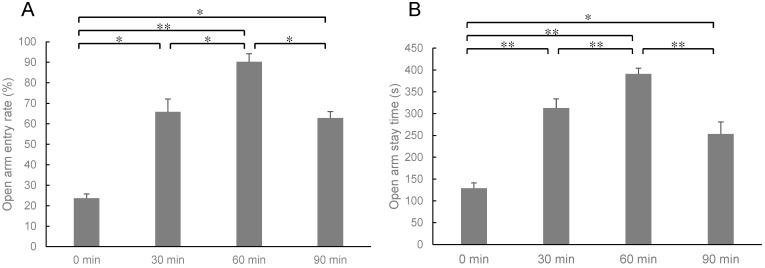
Elevated plus-maze test. (**A**) Open arm entry rate (%), (**B**) open arm stay time (s). Results represent mean ± SEM (n = 6). * *p* < 0.05, ** *p* < 0.01 (Two-way ANOVA followed by Tukey’s test or Student’s *t*-test).

**Figure 3 molecules-27-02661-f003:**
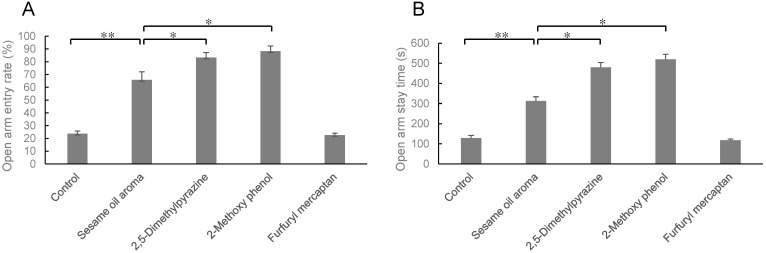
Elevated plus-maze test. (**A**): open arm entry rate (%), (**B**): open arm stay time (s). Results represent mean ± SEM (n = 6). * *p* < 0.05, ** *p* < 0.01 (Two-way ANOVA followed by Tukey’s test or Student’s *t*-test).

**Figure 4 molecules-27-02661-f004:**
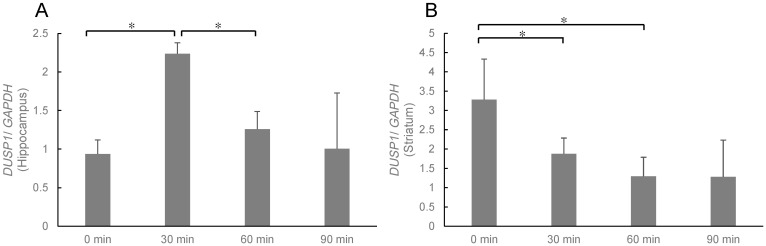
Expression level of *DUSP1* in the hippocampus (**A**) and striatum (**B**). Results show means ± SEM (control/saline, stress/saline, control/sesame, stress/sesame, n = 6). * *p* < 0.05 (two-way ANOVA followed by Tukey’s test or Student’s *t*-test).

**Figure 5 molecules-27-02661-f005:**
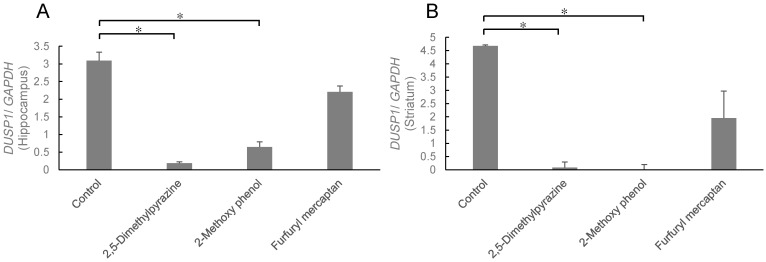
Expression level of *DUSP1* in the hippocampus (**A**) and striatum (**B**). Results show means ± SEM (control/saline, stress/saline, control/sesame, stress/sesame, n = 6). * *p* < 0.05 (two-way ANOVA followed by Tukey’s test or Student’s *t*-test).

**Table 1 molecules-27-02661-t001:** The primer sequence of each gene measured in the present study.

*GAPDH*	Forward: 5′-GGGTCCCAGCTTAGGTTCATCA-3′
	Reverse: 5′-GTTCACACCCACCTTCACCATT-3′
*DUSP1*	Forward: 5′-CGCAGTGCCTGTTGTTGGA-3′
	Reverse: 5′-TGAAGCGCACGTTCACTGAG-3′

## Data Availability

The data presented in this study are available on request from the corresponding author.

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
