# Peer review of "Stress-Relieving Effects of Sesame Oil Aroma and Identification of the Active Components"

_molecules, 2022, doi:10.3390/molecules27092661_

Round 1

Reviewer 1 Report

In this study, Takemoto and colleagues evaluated the stress relieving effects of sesame oil aroma in mice and addressed the active components present in it. Despite of the general interest of the work, there are some major aspects that should be carefully addressed:

- first of all, it is not clear whether all the active components were identified, as the authors only gave emphasis to the 3 main compounds, but there are also some others that were not mentioned here.

- is the methodology used the most reliable to address the stress relieving effects of such type of matrices? No refs were added in the materials and methods section

- what about the toxicological profile of the studied matrix? What is the pharmacokinetics and pharmacodynamics of the isolated compounds studied?

- English language should be carefully revised

Specific comments

- l. 40: sleep disorders are not an example of neuropsychiatric disorder; please clarify

- l. 63-72: please make a slight mention to the main compounds present in such matrices, also indicating the variations in abundance depending on the geographical conditions, that also play a major role

- l. 100: revise the sentence

- Figure 5: why the effects of the oil were not addressed?

- l. 159: “high concentrations” > which concentrations? Please detail

- l. 216-218: remove the sentence

- l. 221: what is the purity and origin of the aromatic sesame oil?

- l. 227-285: please support with proper refs the materials and methods section

- l. 287-293: what was the statistical software used for analysis? Also, delete lines 291-293

- l. 295-304: a mention to the toxicological profile of the matrix, as well as its bioavailability and safety features should be done. The same for the individual compounds that revealed a good potential

Author Response

Thank you very much for your valuable comments. Based on your advice, we have improved the manuscript as follows, and revised part was yellow highlighted. And the answers to your comments were also described as follows.

  1. first of all, it is not clear whether all the active components were identified, as the authors only gave emphasis to the 3 main compounds, but there are also some others that were not mentioned here.

This experiment focused on the major compounds found in sesame oil (2,5-dimethylpyrazine, 2-methoxy phenol and furfuryl mercaptan). Especially, it is reported that alkylpyrazine possesses GABAnergic effect (Line 204-207). Therefore, these compounds were the focus of this experiment. Evaluation of the activity of other components will be the subject of future work.

  1. is the methodology used the most reliable to address the stress relieving effects of such type of matrices? No refs were added in the materials and methods section

Water immersion stress was applied to mice, which is thought to be suitable for preparing model animals before development of depression (undiagnosed condition). We previously demonstrated alterations in the expression of genes and proteins in the brain after stress, and suggested that the aroma of coffee beans has anti-stress effects on the brain.

  1. what about the toxicological profile of the studied matrix? What is the pharmacokinetics and pharmacodynamics of the isolated compounds studied?

The safety of sesame rignans is described (Line 223-229). Safety of sesame aroma fragrances has not been reported, and there are also no reports on pharmacokinetics.

  1. English language should be carefully revised

I will ask about English proofreading to MDPI office.

  1. - l. 40: sleep disorders are not an example of neuropsychiatric disorder; please clarify

The manuscript was revised (Line 45-46).

  1. - l. 63-72: please make a slight mention to the main compounds present in such matrices, also indicating the variations in abundance depending on the geographical conditions, that also play a major role

Major fragrant compounds contained in sesame oil was described in Line 78-81. There are no reports about seasonal change of fragrant components in sesame oil aroma.

  1. - l. 100: revise the sentence

The Figures (Fig. 2~5) were revised: stress control→0 min or control.

  1. - Figure 5: why the effects of the oil were not addressed?

DUSP1 expression after inhalation of sesame oil aroma was described in Figure 4.

  1. - l. 159: “high concentrations” > which concentrations? Please detail

The manuscript was revised (Line 169-171).

  1. - l. 216-218: remove the sentence

We removed the pointed sentence.

  1. - l. 221: what is the purity and origin of the aromatic sesame oil?

The sesame oil used in this study was purchased from Kuki Sangyo, and this oil was produced by pressing and extracting from white sesame from Guatemala, followed by a cooling process. And this oil is not a mixture.

  1. - l. 227-285: please support with proper refs the materials and methods section

The experimental method is almost the same as in the previous report. References were cited for each method.

  1. - l. 287-293: what was the statistical software used for analysis? Also, delete lines 291-293

Analyses were performed with BellCurve for Excel, and we revised the manuscript (Line 311-313).

Reviewer 2 Report

L-31 Stress is one of the most prevalent psychological disorders; please include some data.

L-46 (...) some of them being used as agents to relieve anxiety and stress - Cite some examples.

L-74 Please explain "(...) and toxic effects (...)" and its relation with L-156 " Increased blood levels of the active ingredient may also be associated with increased efficacy" ¿and toxic effects?

L-177, Is that a problem? Please cite possible pharmaceutical technology that could solve that problem.

Author Response

Thank you very much for your valuable comments. Based on your advice, we have improved the manuscript as follows, and revised part was yellow highlighted. And the answers to your comments were also described as follows.

  1. L-31 Stress is one of the most prevalent psychological disorders; please include some data.

The explanation of stress disorder was added in Line 33-39.

  1. L-46 (...) some of them being used as agents to relieve anxiety and stress - Cite some examples.

Some examples were added in Line 53-55.

  1. L-74 Please explain "(...) and toxic effects (...)" and its relation with L-156 " Increased blood levels of the active ingredient may also be associated with increased efficacy" ¿and toxic effects?

The main effects and side effects of inhalation administration are discussed in Line 171-177.

  1. L-177, Is that a problem? Please cite possible pharmaceutical technology that could solve that problem.

Consideration has been changed (Line 208-222).

Reviewer 3 Report

This study aimed to analyze the stress-relieving properties of sesame essential and three of its active ingredients. The authors may address the following issues to strengthen the manuscript:

  • Line 153 y 154 should be modified. The authors said that the “Anti-stress effect of sesame oil aroma appeared 30 min after inhalation.”, but we are not sure that the effect appeared before 30 min.
  • The authors use a piece of filter paper impregnated with 50 μl of sesame oil. Without dilution?
  • It could be interesting if the authors measured the amount of essential oil evaporated in a cage for an hour for each concentration. Check the reference Journal of Traditional and Complementary Medicine, 2016, 6(2), 140-145.

Author Response

Thank you very much for your valuable comments. Based on your advice, we have improved the manuscript as follows, and revised part was yellow highlighted. And the answers to your comments were also described as follows.

  1. Line 153 y 154 should be modified. The authors said that the “Anti-stress effect of sesame oil aroma appeared 30 min after inhalation.”, but we are not sure that the effect appeared before 30 min.

The sentence was revised as Line 163-164.

  1. The authors use a piece of filter paper impregnated with 50 μl of sesame oil. Without dilution?

We used sesame oil without dilution.

  1. It could be interesting if the authors measured the amount of essential oil evaporated in a cage for an hour for each concentration. Check the reference Journal of Traditional and Complementary Medicine, 2016, 6(2), 140-145.

We will try to measure the concentration of fragrance in the future.

Round 2

Reviewer 1 Report

Accept